# ConsisDrive: Identity-Preserving Driving World Models for Video Generation by Instance Mask

**Zhuoran Yang, Yanyong Zhang** [*]
University of Science and Technology of China
shanpoyang@mail.ustc.edu.cn, yanyongz@ustc.edu.cn

## Abstract

Autonomous driving relies on robust models trained on large-scale, high-quality multi-view driving videos. Although world models provide a cost-effective solution for generating realistic driving data, they often suffer from identity drift, where the same object changes its appearance or category across frames due to the absence of instance-level temporal constraints. We introduce **ConsisDrive**, an identity-preserving driving world model designed to enforce temporal consistency at the instance level. Our framework incorporates two key components: (1) Instance-Masked Attention, which applies instance identity masks and trajectory masks within attention blocks to ensure that visual tokens interact only with their corresponding instance features across spatial and temporal dimensions, thereby preserving object identity consistency; and (2) Instance-Masked Loss, which adaptively emphasizes foreground regions with probabilistic instance masking, reducing background noise while maintaining overall scene fidelity. By integrating these mechanisms, ConsisDrive achieves state-of-the-art driving video generation quality and demonstrates significant improvements in downstream autonomous driving tasks on the nuScenes dataset. Our project page is here[1].

## 1 Introduction

Autonomous driving has attracted extensive attention from both academia and industry over the past decades Shi et al. (2016); Zheng et al. (2024a); Chen et al. (2024); Jiang et al. (2023). To achieve reliable performance, autonomous systems rely on high-quality, large-scale multi-view driving videos with precise annotations, which are essential for training perception, tracking, and planning models. However, collecting and labeling such real-world driving data is both costly and labor-intensive. Benefiting from the rapid advancements in generative video models Lei et al. (2023); Xi et al. (2025); Zheng et al. (2024b); HaCohen et al. (2024); Wang et al. (2024a); Gao et al. (2024a); Zhou et al. (2024); Ho et al. (2022); Blattmann et al. (2023); Hu (2024); Wang et al. (2023a; 2024b); Bar-Tal et al. (2024); Gupta et al. (2024), driving world models Zhao et al. (2024); Wen et al. (2024); Jia et al. (2023); Wang et al. (2023c); Gao et al. (2024b) have emerged as a promising alternative. These models can synthesize diverse and realistic driving scenarios at scale, significantly reducing the demand for costly data collection and annotation.

Instance identity preservation across frames is critical for generating realistic driving videos, as it directly affects video quality Unterthiner et al. (2018) and determines their applicability in downstream autonomous driving tasks. For example, multi-object tracking Wang et al. (2023b) and perception tasks Wang et al. (2023b) require temporally stable instance appearances to ensure reliable temporal context understanding. Similarly, planning Hu et al. (2023) relies on temporally coherent trajectories of surrounding agents to support accurate motion forecasting. These requirements necessitate that the world model consistently maintains instance identities—such as category, color, and shape—across consecutive frames, ensuring continuity in both appearance and behavior of dynamic objects. From a broader perspective, the ability to preserve instance identity is essential for

---

[*]Corresponding Author
[1]https://shanpoyang654.github.io/ConsisDrive/page.html

world models to effectively capture the underlying dynamics of real-world environments. Technically, enforcing strong temporal consistency improves the reliability of autonomous driving models trained on synthetic data, ultimately enhancing their generalization to real-world scenarios.

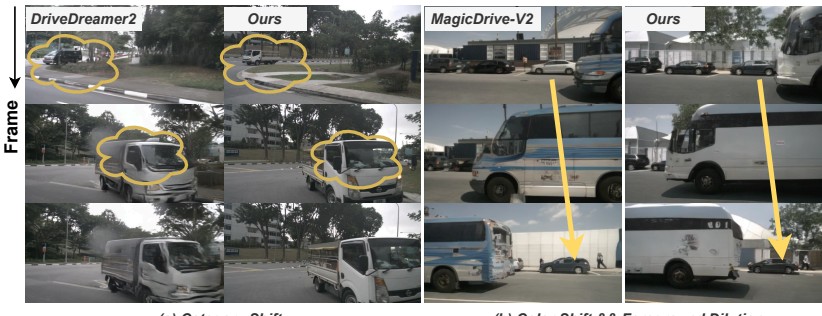

Figure 1: **Limitations of Prior Works in Instance Identity Preservation Across Frames. (a)** *Category Shift*: In DriveDreamer2 Zhao et al. (2024), the bus gradually turns into a truck, indicating a failure to preserve semantic identity over time. **(b)** *Color Shift*: In MagicDrive-V2 Gao et al. (2024b), the car's color changes inconsistently across frames, violating temporal appearance consistency. **(b)** *Foreground Dilution*: In MagicDrive-V2 Gao et al. (2024b), scene-level supervision dilutes supervision over critical foreground regions, breaking temporal identity consistency for small instances like pedestrians. In contrast, our method explicitly enforces instance-level temporal constraints, maintaining consistency across frames and effectively addressing these issues.

However, existing diffusion-based world models frequently suffer from *identity drift*, where the same object changes its appearance or even category across frames (e.g., a red car becoming black, or a bus turning into a truck), as shown in Fig. 1. Such identity inconsistency severely degrades video realism and limits the applicability of generated data for downstream driving tasks. We identify three major root causes of this problem. First, the absence of explicit instance identity conditions prevents the model from anchoring consistent identities over long horizons. For example, DriveDreamer2 Zhao et al. (2024) does not incorporate instance-specific conditions such as category, leading to noticeable semantic shifts, as illustrated in Fig. 1(a). This highlights the necessity of injecting explicit instance identity signals into the generation process. Second, the attention mechanism of current diffusion transformers is not instance-aware. For instance, FLUX's MMDiT Labs (2024) computes 3D full attention across all visual tokens from different instances. This makes the attention mechanism unreliable and causes information leakage between different instances. Models such as MagicDrive-V2 Gao et al. (2024b) integrate temporal attention layers to enhance inter-frame *global* coherence, they lack fine-grained, instance-aware temporal alignment, suffering from identity inconsistencies such as color shifts, as shown in Fig. 1(b). This underscores the need for instance-aware attention mechanisms. Third, existing training objectives Wen et al. (2024) apply uniform supervision over the entire frame, forcing the model to reconstruct both background pixels (e.g., sky, buildings) and critical foreground regions with equal importance. Since background pixels dominate the scene, this *supervision dilution* prevents the model from focusing on fine-grained identity-preserving features. Consequently, foreground temporal consistency is easily broken, especially for small objects, as shown in Fig. 1(b) in MagicDrive-V2 Gao et al. (2024b). This motivates the design of instance-aware training objectives that emphasize foreground regions.

To address the above challenges, we propose **ConsisDrive**, an identity-preserving driving world model specifically designed to enforce instance-level temporal consistency. Our framework incorporates instance awareness into both the attention mechanism and the training objective, guided by carefully constructed instance masks. In particular, we introduce two core components: The **I**nstance **M**asked **A**ttention (**IMA**) module explicitly guides the model's attention towards each individual instance, effectively preventing information leakage across multiple instances. Specifically, by constructing *instance identity mask*, we restrict visual tokens to attend only to the identity embeddings of their corresponding instances. This preserves the identity of the instance across long sequences, effectively mitigating identity drift (e.g., preventing a bus from gradually being interpreted as a truck). What's more, by constructing *instance trajectory mask*, we ensure that tokens of the same object across frames exclusively attend to each other, while interactions across different

instances are strictly blocked. This design allows the model to reliably propagate appearance features such as color and texture along the trajectory of each object, thereby avoiding cross-instance information leakage and ensuring consistent instance-level visual identity across time. Second, the **Instance Masked Loss (IML)** addresses the supervision dilution problem caused by uniform loss computation across entire frames. IML employs instance masks to emphasize supervision on foreground regions during training. A probabilistic dynamic masking strategy is further introduced, which adaptively balances between foreground-focused loss and global reconstruction loss. This design ensures that foreground consistency is enforced without sacrificing overall scene fidelity, allowing the model to capture both fine-grained identity details and natural background appearance.

Through the joint design of Instance Masked Attention module and Instance Masked Loss Supervision, ConsisDrive significantly mitigates identity drift and achieves temporally consistent video generation for driving scenarios. Our approach achieves state-of-the-art performance in both video generation quality and downstream autonomous driving task validation, outperforming previous works Gao et al. (2024b); Zhao et al. (2024); Li et al. (2023); Wen et al. (2024). Our contributions are as follows.

- We propose the Instance-Masked Attention module, which explicitly directs the model's attention to each individual instance. By incorporating both an instance identity mask and a trajectory mask, the module constrains visual tokens to attend exclusively to tokens of their corresponding instances across spatial and temporal dimensions. This design effectively enforces instance-level temporal consistency while preventing information leakage between different instances.
- We design Instance-Masked Loss Supervision, a probabilistic instance-focused training objective that employs instance masks to emphasize supervision on foreground regions.
- Our model achieves SOTA video generation quality with high FID and FVD on the nuScenes benchmark, surpassing previous methods. For autonomous driving applications, the generated videos are validated on downstream perception, tracking, and planning tasks, with performance competitive to real-world sensor data.

## 2 RELATED WORKS

**Street-View Generation.** Street-view generation methods typically use 2D layouts like BEV maps, 2D bounding boxes, and semantic segmentation. BEVGen Swerdlow et al. (2023) encodes semantic data in BEV layouts, while BEVControl Yang et al. (2023) uses a two-stage pipeline for multi-view urban scenes, ensuring cross-view consistency. However, projecting 3D information into 2D layouts loses geometric details, causing temporal inconsistencies in videos. To address this, we use 3D bounding boxes to maintain geometric fidelity. Unlike DrivingDiffusion Li et al. (2023), which relies on a complex multi-stage pipeline, our method simplifies the process with an efficient, end-to-end framework, ensuring temporal coherence and computational efficiency.

**Simulation-to-Real Visual Translation.** Recent advances in synthetic data for real-world visual tasks have shown significant progress. GAN-based translation Guo et al. (2020) and domain randomization Tobin et al. (2017) bridge synthetic and real-world data distributions, while datasets like Synthia Ros et al. (2016) and Virtual KITTI Cabon et al. (2020) provide scalable benchmarks for semantic segmentation and autonomous driving. Adversarial training Shrivastava et al. (2017); Zhang et al. (2018) reduces distribution gaps, and human motion representation learning Guo et al. (2022) highlights synthetic data's utility in video understanding. Unlike these methods, we extract proxy data like 3D bounding boxes and road maps from graphics systems, leveraging these conditions to generate more realistic and diverse videos.

## 3 METHOD

We introduce ***ConsisDrive***, an identity-preserving driving world model specifically designed to enforce instance-level temporal consistency. Our framework incorporates instance awareness into both the 3D full attention mechanism (Sec. 3.2) and the training objective (Sec. 3.3), where carefully constructed instance masks serve as structural priors to guide consistency across frames. The overall architecture of our model is illustrated in Fig. 2.

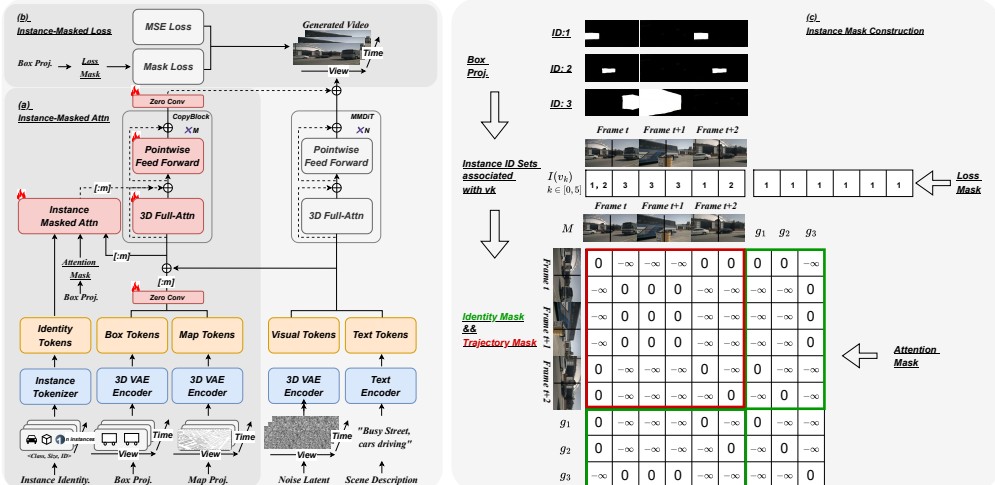

Figure 2: **Overview.** (a) Instance-Masked Attention, which explicitly directs the model's attention to each individual instance by incorporating both an instance identity mask and trajectory mask. (b) Instance-Masked Loss Supervision, a probabilistic instance-focused training objective that employs instance loss masks to emphasize supervision on foreground regions. (c) Instance Mask Construction. Illustration of how the Instance Identity Mask, Instance Trajectory Mask, and Instance Loss Mask are constructed from 3D box projections.

## 3.1 OVERVIEW

Building on OpenSora V2.0 Peng et al. (2025), we employ a Variational Auto-Encoder (Video DC-AE) for video encoding, T5XXL Chung et al. (2024) and CLIP-Large Radford et al. (2021) for text encoding, MMDiT Labs (2024) as the foundational model for the denoising process.

To achieve fine-grained control, we introduce a comprehensive set of control conditions, including bounding box projection, road maps, and scene descriptions, integrating them into the conditioned video generation process. Moreover, we introduce instance category label, instance size, and tracking ID to extract instance identity conditions and incorporate them into the attention process, a key mechanism within our proposed Instance-Masked Attention module, detailed in Sec. 3.2.

Given the need to handle multiple control elements, we adopt ControlNet Zhang et al. (2023) to inject control signals into the video generation process. To integrate these control-aware representations, we duplicate the first 19 base blocks of the double-stream MMDiT backbone as dedicated control blocks. Each control block fuses condition features with the corresponding outputs of the base blocks, thereby modulating the feature flow and ensuring that control signals are effectively incorporated throughout the generation pipeline.

## 3.2 INSTANCE-MASKED ATTENTION

The Instance-Masked Attention module, as illustrated in Fig.2 (a), is designed to guide the model's attention towards each individual instance. By constructing instance masks from the bounding boxes of objects in Fig.2 (c), we effectively prevent information leakage across multiple instances. We add our proposed learnable Instance-Masked Attention module to handle the per-instance identity conditioning. Instance-Masked Attention module fuses the instance identity conditions with the corresponding outputs of the copy blocks and modulates its features to enable instance-aware attention. We now describe the key operations within the Instance-Masked Attention module in detail.

The Instance-Masked Attention module, as illustrated in Fig. 2(a), is designed to explicitly guide the model's attention towards each individual instance. Instance masks are constructed from object bounding boxes, as shown in Fig. 2(c), to prevent information leakage across instances and ensure instance-level disentanglement. We introduce a learnable Instance-Masked Attention mechanism, which injects per-instance identity conditions and fuses them with the corresponding outputs of the copy blocks. By modulating these features, the module enables instance-aware attention that

preserves instance identity and ensures instance consistency across spatial and temporal dimensions. We now describe the key operations of the Instance-Masked Attention module in detail.

### 3.2.1 INSTANCE IDENTITY CONDITION

For each instance $i$ that appears in the video, we construct a global condition that integrates multiple factors of the instance, including its category, its unique tracking ID, and the size of its bounding box. Together, these attributes provide an informative representation of both semantic identity and geometric configuration, which is crucial for preserving the instance identity consistently across frames.

Concretely, we first apply a Fourier mapping $\gamma(\cdot)$ to encode the tracking ID $ID_i$ and the bounding box size $s_i = (dx_i, dy_i, dz_i)$. At the same time, we employ the CLIP-Large Radford et al. (2021) text encoder $\tau_\theta(\cdot)$ to extract a semantic feature from the category label $c_i$. These components are concatenated and passed through a multilayer perceptron (MLP) to produce the instance global identity condition embedding:

$$g_i = \text{MLP}([\tau_\theta(c_i), \gamma(s_i), \gamma(ID_i)]). \tag{1}$$

The complete set of embeddings for all $n$ instances in the video is then denoted as:

$$G = \{g_i\}_{i=1}^n.$$

### 3.2.2 INSTANCE-MASKED ATTENTION AND FUSE MECHANISM

We denote the $m = T_{compress} \times H_{compress} \times W_{compress}$ visual tokens extracted from the copy MMDiT block as $V$, and the $n$ instance condition tokens as $G$. We then apply 3D self-attention (SA) over the backbone features and the concatenated instance condition tokens $[V, G]$ in the Instance-Masked Attention module, which can be formulated as:

$$\tilde{V} = \text{SA}_{\text{mask}}([V, G]). \tag{2}$$

We observed that standard self-attention, without masking, leads to information leakage across instances, such as the color of one instance bleeding into another. To address this problem, we construct a mask matrix $M \in \mathbb{R}^{(m+n) \times (m+n)}$ to determine the valid connections:

**Token-to-Instance Indicator Function.** Indicator function $I(v_k)$ denotes the set of instance IDs whose projected bounding boxes cover the spatial region of token $v_k$, as illustrated in Fig.2 (c). Formally, each instance $i$ is represented by a 3D bounding box with corners $C_i = \{\mathbf{X}_{i,c}\}_{c=1}^8$, where $\mathbf{X}_{i,c} \in \mathbb{R}^3$. Given camera parameters $(\mathbf{K}^t, \mathbf{R}^t, \mathbf{T}^t)$ at frame $t$, the corners are projected as

$$\tilde{\mathbf{x}}_{i,c}^t = \mathbf{K}^t \left(\mathbf{R}^t \mathbf{X}_{i,c} + \mathbf{T}^t\right), \quad \mathbf{x}_{i,c}^t = \left(\tfrac{\tilde{x}}{\tilde{z}}, \tfrac{\tilde{y}}{\tilde{z}}\right), \tag{3}$$

with the convex hull of $\{\mathbf{x}_{i,c}^t\}$ forming the polygon $P_i^t$. Rasterization yields binary masks $BM_i \in \{0,1\}^{T \times H \times W}$. We further apply trilinear interpolation to map these masks into latent space, denoted as $\tilde{BM}_i$. For patch tokens $p$ obtained from VAE compression, we define

$$I(v_k) \equiv I(t, p) = \{ i \mid \exists (x, y), \ \tilde{BM}_i(t, x, y) = 1 \}, \tag{4}$$

where $v_k$ is a flattened visual token corresponding to $(t, p)$.

**Instance Identity Mask.** For each visual token $v_k$ and instance condition token $g_i$, the attention score is masked as

$$M_{k,m+i}/M_{m+i,k} = -\infty \quad \text{if } i \notin I(v_k),$$

This mask ensures that each visual token can only attend to the identity condition token of the corresponding instance. This mechanism explicitly injects global identity features into the attention process, ensuring that each instance preserves its identity consistently across long temporal sequences. Meanwhile, it strictly suppresses interactions between different instances, thereby preventing identity leakage across objects.

**Instance Trajectory Mask.** For two visual tokens $v_k$ and $v_j$, the attention score is masked as

$$M_{k,j} = -\infty \quad \text{if } I(v_k) \cap I(v_j) = \emptyset.$$

This mask ensures that tokens of the same instance across frames can attend to one another, while interactions across different instances are strictly prohibited. This design preserves temporal consistency by explicitly enabling the propagation of instance-specific features across their trajectories.

Finally, the output of the Instance-Masked Attention is added back to the backbone representation via gated addition:

$$V = V + \tanh(\omega)\, \tilde{V}[:m], \tag{5}$$

where $\omega$ is a learnable scalar parameter, initialized to zero, that adaptively controls the contribution of the Instance-Masked Attention module.

### 3.3 INSTANCE-MASKED LOSS

The goal of ***ConsisDrive*** is to ensure that foreground objects in driving scenes remain consistent across all frames in the generated video. However, existing training objectives apply uniform supervision over the entire frame, forcing the model to reconstruct both background pixels (e.g., sky, buildings) and critical foreground regions with equal importance. Since background pixels dominate the scene, this supervision dilution introduces noise that interferes with model training, preventing the model from focusing on fine-grained identity-preserving features. Consequently, foreground temporal consistency is easily broken, especially for small objects.

To resolve this issue, we propose the *Instance-Masked Loss*, which focuses the supervision signal on foreground objects, as illustrated in Fig.2 (b)(c). Specifically, we construct a binary loss mask $M_{\text{Loss}} \in \{0,1\}^{T_{\text{comp}} \times H_{\text{comp}} \times W_{\text{comp}}}$ from the Instance-to-Token Indicator function $I(v_k)$. For each token $v_k$, the mask is defined as $M_{\text{Loss}}(v_k) = 1_{\{I(v_k) \neq \varnothing\}}$, ensuring that only tokens covered by at least one instance are selected. The masked loss is then computed as:

$$L_{\text{mask}} = M_{\text{Loss}} \odot L,$$

where $L$ denotes the original denoising loss and $\odot$ indicates element-wise multiplication.

However, directly applying this masked loss to all training samples may cause the model to overfit to foreground objects, which in turn harms the generation quality of background regions, such as roads and high-definition maps. To alleviate this problem, we adopt a probabilistic dynamic masking strategy. Specifically, with a probability $p$ of $\alpha$, the masked loss is applied:

$$\tilde{\mathcal{L}}_{\text{mask}} = \begin{cases} \mathcal{L}_{\text{mask}}, & \text{if } p < \alpha \\ \mathcal{L}, & \text{if } p \geq \alpha \end{cases} \tag{6}$$

This stochastic scheme allows the model to concentrate on foreground consistency while still preserving the natural realism of background content.

## 4 EXPERIMENT

### 4.1 SETUPS

**Datasets and Baselines.** We train and evaluate our model on the nuScenes dataset Caesar et al. (2020). To benchmark our approach, we compare it with state-of-the-art driving world models, including BEVControl Yang et al. (2023), DriveDiffusion Li et al. (2023), DriveDreamer2 Zhao et al. (2024), Panacea Wen et al. (2024), and MagicDrive-V2 Gao et al. (2024b).

**Metrics.** For realism assessment, we use FID Heusel et al. (2017) and FVD Unterthiner et al. (2018) to measure video quality. To evaluate the effectiveness of attribute binding and Instance Masked Loss Supervision, we measure the alignment between generated instances and their conditioned categories and sizes, ensuring that both semantic and geometric structures are faithfully preserved. This evaluation is conducted through perception tasks, since accurate category recognition and size localization are fundamental requirements for reliable perception. Hence, perception performance directly reflects the generation accuracy of object categories and spatial extents. Specifically, following Panacea Wen et al. (2024), we adopt the video-based perception model StreamPETR Wang et al. (2023b) and report metrics such as the nuScenes Detection Score (NDS) and mean Average Precision (mAP). Among them, mAP directly measures the accuracy of object category detection, while NDS integrates category detection with localization, orientation, and other aspects to provide

a holistic assessment of perception quality. To further evaluate the propagation of instance-specific features across frames, we assess our model on the multi-object tracking (MOT) task in real-world autonomous driving scenarios. MOT explicitly measures the ability to maintain consistent object identities over time, using metrics like ID switches (IDS). We also adopt the StreamPETR model Wang et al. (2023b) as the tracker and report standard MOT metrics, including AMOTA, AMOTP, and IDS. Although Bevfusion Liu et al. (2023) has also been employed for perception evaluation Gao et al. (2024c), it is based on an image model, performs worse than the video-based StreamPETR, and lacks the ability to provide object tracking metrics. For these reasons, we choose StreamPETR as our evaluation model.

| Method | Multi-View | Multi-Frame | FVD↓ | FID↓ |
|---|---|---|---|---|
| BEVControl Yang et al. (2023) | ✓ | | - | 24.85 |
| DrivingDiffusion Li et al. (2023) | ✓ | ✓ | 332 | 15.83 |
| Panacea Wen et al. (2024) | ✓ | ✓ | 139 | 16.96 |
| MagicDrive-V2 Gao et al. (2024b) | ✓ | ✓ | 94.84 | 20.91 |
| DriveScape Wu et al. (2024) | ✓ | ✓ | 76.39 | 8.34 |
| DriveDreamer2 Zhao et al. (2024) | ✓ | ✓ | 55.7 | 11.2 |
| DiVE Jiang et al. (2024) | ✓ | ✓ | 94.6 | - |
| DrivingSphere Yan et al. (2024) | ✓ | ✓ | 103.42 | - |
| UniScene Li et al. (2025) | ✓ | ✓ | 70.52 | 6.12 |
| InstaDrive Yang et al. (2025) | ✓ | ✓ | 38.06 | 3.96 |
| UniMLVG Chen et al. (2025) | ✓ | ✓ | 60.1 | 8.8 |
| *ConsisDrive* | ✓ | ✓ | 37.23 | 3.88 |

Table 1: Visual and Temporal Fidelity: Comparison with SoTA methods on nuScenes validation set.

## 4.2 TRAINING DETAILS

Our method is implemented based on OpenSora V2.0 Peng et al. (2025). All training inputs were set to 16x256x448 and conducted on $64$ A100 GPUs. Experimental results show that our method can stably generate over 200 frames.

## 4.3 MAIN RESULTS

### 4.3.1 QUANTITATIVE ANALYSIS

To verify the fine-grained temporal consistency and high fidelity of our generated videos, we compare our approach with various state-of-the-art driving world models. We generate training and validation data using the nuScenes dataset's labels as conditions.

**Visual Realism and Temporal Fidelity.** Our generated videos achieve superior temporal fidelity, reducing the FVD to $37.23$. This improvement stems from the proposed **Instance-Masked Attention** module, which preserves instance attributes across frames and thereby enhances instance-level temporal consistency. In terms of visual quality, our method attains an FID of $3.88$, as reported in Tab. 1, substantially outperforming both video-based approaches (e.g., DriveDreamer2) and image-based solutions (e.g., BEVControl). These demonstrate that our generated images exhibit not only stronger temporal coherence but also significantly higher visual realism.

| Method | Real | Gen. | mAP↑ | mAOE↓ | mAVE↓ | NDS↑ |
|---|---|---|---|---|---|---|
| Oracle | ✓ | - | 34.5 | 59.4 | 29.1 | 46.9 |
| Panacea | - | ✓ | 22.5 (65.22%) | 72.7 | 46.9 | 36.1 (76.97%) |
| Panacea | ✓ | ✓ | 37.1 (+2.6%) | 54.2 | 27.3 | 49.2 (+2.3%) |
| *ConsisDrive* (Ours) | - | ✓ | 31.5 (91.3%) | 63.0 | 33.1 | 42.06 (89.68%) |
| *ConsisDrive* (Ours) | ✓ | ✓ | 43.2 (+8.7%) | 39.8 | 25.2 | 54.6 (+7.7%) |

Table 2: Comparison on perception tasks with Panacea. Training StreamPETR with synthetic data augmentation leads to significant performance improvements, highlighting the value of generated data for perception.

| Method | Real | Gen. | NDS↑ |
|---|---|---|---|
| Oracle | ✓ | - | 46.90 |
| Panacea | - | ✓ | 32.10 (68.00%) |
| MagicDrive-V2 | - | ✓ | 36.82 (78.51%) |
| *ConsisDrive* | - | ✓ | 41.38 (88.23%) |

Table 3: Comparison of perception task performance on generated nuScenes (T+I)2V validation data. Evaluated with pre-trained StreamPETR Wang et al. (2023b), our model outperforms baselines without post-refinement, showing its ability to faithfully capture and bind instance attributes.

| Method | Real | Gen. | AMOTA↑ | AMOTP↓ | IDS↓ |
|---|---|---|---|---|---|
| Oracle | ✓ | - | 0.289 | 1.419 | 687 |
| DriveDreamer2 | ✓ | ✓ | 0.313 | 1.387 | 593 (-94) |
| InstaDrive | ✓ | ✓ | 0.496 | 1.376 | 532 (-155) |
| *ConsisDrive* (Ours) | ✓ | ✓ | 0.498 | 1.350 | 525 (-162) |

Table 4: Comparison involving data augmentation using synthetic data on multi object tracking.

**Instance Attribute Binding.** We evaluate instance-level temporal consistency by examining how well global instance attributes are bound to the generated content. In autonomous driving, perception tasks critically depend on precise category recognition and size localization, making them a natural measure of attribute binding quality. To this end, we assess the alignment between generated instances and their conditioned categories and sizes, which directly reflects the fidelity of instance attribute binding.

*Data Augmentation Performance on Perception.* As shown in Tab. 2, training StreamPETR exclusively on our generated dataset achieves a mean Average Precision (mAP) of 31.5%, which corresponds to 91.3% of the performance obtained by training solely on the real nuScenes dataset. Since mAP directly measures category detection accuracy, this result confirms that global instance attributes are reliably bound through our Instance Identity Mask (Sec. 3.2). Moreover, the generated dataset proves to be not only a viable substitute for real data but also an effective standalone training resource for perception models. When we further re-train StreamPETR by augmenting real data with our generated videos, the perception model achieves a nuScenes Detection Score (NDS) of 54.6, representing a 7.7-point improvement over training with real data alone. This demonstrates the substantial benefit of incorporating our generated data into the training pipeline.

*Validation Performance on Perception.* Additionally, we use the pre-trained StreamPETR model to evaluate the generated validation set of nuScenes. As reported in Tab. 3, our model achieves a relative performance of 88.23% on the NDS metric, highlighting strong alignment between generated content and conditioned instance categories and sizes.

**Instance Attribute Propagation.** *Data Augmentation Performance on MOT.* We further evaluate instance-level temporal consistency by measuring how effectively instance attributes are propagated across frames. In autonomous driving, this is naturally reflected by the multi-object tracking (MOT) task, since MOT requires tracking instances according to their consistent attributes over time. This makes MOT a strong indicator of attribute propagation quality. Concretely, we generate videos conditioned on labels from the nuScenes training set and use them to augment the training data and re-train the object tracking model StreamPETR Wang et al. (2023b). Then we evaluate the real validation set using the re-trained StreamPETR. As shown in Tab.4, the MOT model re-trained with our generated data shows significant improvements, achieving a lower number of identity switches (IDS = 525) compared to the original pre-trained StreamPETR. This demonstrates that the Instance Trajectory Mask in Instance-Masked Attention module effectively propagates and preserves instance attributes, ensuring reliable temporal consistency for downstream perception tasks.

### 4.3.2 QUALITATIVE ANALYSIS

We qualitatively compare *ConsisDrive* against state-of-the-art baselines by inspecting generated videos, as illustrated in Fig. 1. Please refer to the project page for additional video result.

**Instance Attribute Binding (Category).** As shown in Fig. 1(a), DriveDreamer2 suffers from semantic drift, where the category of a car gradually changes into a truck. In contrast, *ConsisDrive* preserves the semantic category consistently across all frames, demonstrating stronger instance-level temporal consistency. This improvement arises from the Instance Identity Mask in the Instance Masked Attention Module, which explicitly binds identity features to their corresponding instance identities.

**Instance Attribute Propagation (Color).** In Fig. 1(b), MagicDrive-V2 exhibits color inconsistency, where the color of the same car changes from black to red across frames. Our model maintains stable color and texture attributes throughout the video, indicating superior attribute propagation over time. This is enabled by the Instance Trajectory Mask in the Instance Masked Attention Module, which enforces cross-frame attention only within the same instance trajectory, preventing appearance drift.

**Foreground Emphasis.** As shown in Fig. 1(b), ***ConsisDrive*** enhances the fidelity of small and challenging objects (e.g., pedestrians), whereas Panacea produces blurred results for such instances. This improvement stems from the Instance Masked Loss Supervision, which prioritizes supervision on foreground regions even when the foreground spatial proportion is very small and background pixels dominate the scene, thereby preventing foreground signals from being diluted by background pixels.

| Settings | FVD↓ | FID↓ | NDS↑ | IDS↓ |
|---|---|---|---|---|
| *ConsisDrive* | 37.23 | 3.88 | 41.38 | 525 |
| w/o IMA(Identity) | 40.89 (+3.66) | 5.29 (+1.41) | 37.55 (-3.83) | 735 (+210) |
| w/o IMA(Trajectory) | 53.66 (+16.43) | 4.41 (+0.53) | 40.40 (-0.98) | 1074 (+549) |
| w/o IML | 40.19 (+2.96) | 4.24 (+0.36) | 36.85 (-4.53) | 637 (+112) |

Table 5: Ablation study results in (T+I)2V scenarios on the nuScenes validation set.

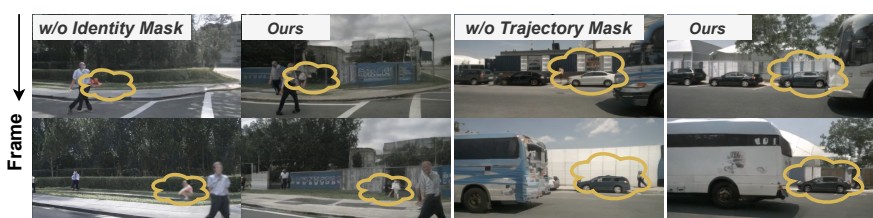

Figure 3: Ablation study of the three key modules. **(a)** Removing the Identity Mask leads to incorrect instance category rendering, e.g., a traffic cone turns into a crouching pedestrian. **(b)** Removing the Trajectory Mask results in color shifts of the car.

## 4.4 ABLATION STUDY

We validate three key components in ***ConsisDrive*** through both qualitative and quantitative analyses, demonstrating their effectiveness and robustness. The qualitative comparison is presented in Fig. 3, while quantitative results are reported in Tab. 5.

**Instance Identity Mask.** To assess the impact of the Instance Identity Mask within the Instance Masked Attention module, we conduct an ablation by removing the global instance identity conditioning (i.e., filling the instance identity mask with $-\infty$). As shown in Tab. 5, the absence of global identity conditioning results in a 3.83-point drop in NDS for the perception task. This confirms its critical role in binding category and size conditions to their corresponding instances, ensuring faithful instance attribute binding.

**Instance Trajectory Mask.** To evaluate the impact of the Instance Trajectory Mask, we perform an ablation by removing trajectory-based attention (i.e., filling the trajectory mask with $-\infty$). As reported in Tab. 5, removing this causes a significant increase of 549 ID switches in the multi-object tracking task and a 16.43-point drop in FVD. These results underscore its importance in propagating and preserving instance attributes such as color consistently across frames.

**Instance Masked Loss Supervision.** To analyze the effect of Instance Masked Loss Supervision, we remove it and retain only the standard denoising loss. As shown in Tab. 5, this results in a 4.53-point degradation in NDS, demonstrating that the module is crucial for emphasizing supervision on foreground regions. By preventing foreground signals from being overwhelmed when their spatial proportion is small, it ensures higher fidelity in generating small and visually challenging objects.

## 5 CONCLUSION

We propose ***ConsisDrive***, an identity-preserving driving world model specially designed to enhance instance-level temporal consistency. Our approach introduces two key advancements: the Instance Masked Attention module, which explicitly directs the model's attention to each individual instance, and the Instance Masked Loss Supervision, which employs instance masks to emphasize supervision on foreground regions. By incorporating these instance-aware mechanisms, ***ConsisDrive*** achieves SOTA generation quality and significantly improves downstream autonomous driving tasks.

ACKNOWLEDGMENTS

This work was supported by the Fundamental and Interdisciplinary Disciplines Breakthrough Plan of the Ministry of Education of China (No. JYB2025XDXM113) and the National Natural Science Foundation of China (No. 62332016).

ETHICS STATEMENT

This work focuses on video generation for autonomous driving research. Our model is trained and evaluated on publicly available datasets (e.g., nuScenes) and is intended solely for academic research. We emphasize that the generated data should not be directly deployed in real-world driving systems without careful validation, as safety-critical applications require rigorous testing.

REPRODUCIBILITY STATEMENT

We provide comprehensive details of the model architecture, training objectives, and evaluation protocols. All datasets used in this work are publicly available, and the implementation details are carefully documented to ensure reproducibility. Code and configuration files are included in the supplementary material to facilitate further research and independent verification.

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
