# OpenReview forum: "ConsisDrive: Identity-Preserving Driving World Models for Video Generation by Instance Mask"
_ICLR.cc/2026/Conference — ICLR 2026 Poster_

### Official Review · Reviewer_V1rg · 2025-10-26

**Soundness:** 2
**Presentation:** 3
**Contribution:** 2
**Rating:** 2
**Confidence:** 5

**Summary:**

This paper presents ConsisDrive, a novel video diffusion model that incorporates instance-specific attention to improve instance consistency in autonomous driving scenarios. To address identity drift, the method introduces an instance global identity condition embedding that represents all instances in the scene. For each instance, specialized attention is applied to patches corresponding to the instance mask, and the output is fused with the standard attention output through gated addition.

**Strengths:**

1. The paper focuses on addressing the important problem of instance consistency in driving scene generation.
2. Experimental results demonstrate that ConsisDrive achieves strong performance on key metrics.

**Weaknesses:**

1. This method adopts full attention, which is very costing
2. Insufficient literature review: The paper does not includerecent driving world models published in late 2024 and early 2025. I list some of them: [UniMLVG](https://arxiv.org/abs/2412.04842), [UniScene](https://arxiv.org/abs/2412.05435), [DrivingSphere](https://arxiv.org/abs/2411.11252), and [DiVE](https://arxiv.org/abs/2409.01595). Notably, UniMLVG and UniScene are open-source, and UniMLVG reports better FVD performance.
3. The current results do not conclusively demonstrate that instance mask attention is the primary contributor to improved instance consistency. The improvement is mainly shown in the quantitative metrics. While Figure 3 suggests the complete pipeline improves instance-level consistency, the compared baselines differ from ConsisDrive in multiple dimensions, including model architecture, pretrained models, and conditioning inputs. The authors should isolate the contribution of the proposed attention in the qualitative results. Additionally, I recommend that the authors provide some challenging cases, such as vehicles that temporarily disappear and later reappear in the scene.

**Questions:**

Please refer to the weaknesses.

---

> ### Author Response · Authors · 2025-11-24
> **Choice of Full Attention, More Baselines, and Non-Continuous Trajectory (See Page 15 in the updated PDF paper for the whole answer which includes new images and tables)**
>
> We sincerely thank Reviewer for your thoughtful feedback and for highlighting the strengths of our work in temporal consistency, experimental validation, and novelty.
> As noted in our original submission, **we have provided a project page with extensive video results included in the supplementary materials. Please ensure that the entire supplementary archive is fully uncompressed before opening the webpage.**
> The page has been successfully tested on both macOS and Windows 10 systems.
> We have made further revisions to the main paper based on the reviewers' feedback. We appreciate your constructive comments and look forward to any additional suggestions.
>
> **Choice of Full Attention in MMDiT**
>
> We appreciate reviewer’s feedback on computational cost of full attention.
> The use of full attention in MMDiT is becoming increasingly dominant, as seen in architectures like OpenSora V2.0, Wan 2.1, and HunyuanVideo. This approach offers three key advantages:
>
> (i) Full attention has demonstrated superior performance by capturing richer spatiotemporal relationships compared to divided 2D+1D spatiotemporal attention.
>
> (ii) It supports unified generation for both images and videos, simplifying training process and improving model scalability.
>
> (iii) It leverages existing LLM-related acceleration capabilities more effectively, enhancing both training and inference efficiency.
>
>
> **More Comparison With Baselines**
>
> We thank the reviewer for pointing out the recent driving world models.
> After carefully reviewing UniMLVG, UniScene, DrivingSphere, and DiVE, we found that their reported performance is lower than ours across multiple metrics, including FID, FVD, and perception-related evaluations.
> We have now included comparisons with these methods and updated Tab.1 accordingly.
>
>
>
> **Ablation Study of Instance-Masked Attention**
>
> We apologize for the typo in Fig.3, where the label "MagicDrive-V2/Panacea" actually referred to our ablation results.
> To clarify, our ablation study isolates the contribution of the Instance-Masked Attention by removing either the Identity Mask or the Trajectory Mask and comparing the results with those of the full model. The typo has been corrected as suggested.
>
>
> **Challenges for Occlusion in Non-Continuous Trajectory**
>
> Method: Instance-Masked Attention ensures consistent instance identity even when objects are occluded and reappear, the principle is demonstrated in Fig.2 (c).
> When an object (e.g., object 1) becomes occluded at frame $t+1$ and reappears at frame $t+2$, the Indicator function $I(v_k)$ tracks the object's presence across frames. At frame $t$, $1 \in I(v_0)$ when the object is visible. At frame $t+1$, the object is occluded, and at frame $t+2$, $1 \in I(v_4)$ when the object reappears.
> To ensure correct feature propagation, we set $M(0, 4) = 0$ in the Instance Trajectory Mask, allowing attention between token $v_0$ and $v_4$. This enables the reliable propagation of instance-specific features across their trajectory, even after occlusion.
>
> Qualitative results demonstrating our method’s handling of non-continuous trajectories are shown in Fig.4, where the car maintains its identity despite temporary occlusion.

---

> > ### Comment · Reviewer_V1rg · 2025-11-24
> >
> > Thanks for the authors' response. Some of my major concerns have been addressed so I plan to increase my score.However, some remaining questions and weaknesses prevent me from fully recommending the manuscript:
> >
> > 1. Though the authors have clarified the typo in Figure 3, it negatively impacted my initial impression of the paper. Moreover, the issues in Figure 3 are not commonly observed in recent literature within the field, at least not in the demos and qualitative results presented in their paper.
> >
> > 2. Figure 4 effectively demonstrates the identity-preservation capability, making me increase my score. I recommend including an example from the nuScenes validation set, where a vehicle appears, disappears, and reappears on the left-hand side. Such an example would enhance the persuasiveness of the paper and its video demonstration.
> >
> > 3. Despite maintaining consistent identity, the generated vehicle shapes is not in high quality. For instance, the bus in Figure 4 and the white car in the last video of the supplementary material exhibit noticeable irregularity in the overal structure. I suggest the authors carefully curate and select more representative, high-quality results to showcase their method's capability.

---

> ### Author Response · Authors · 2025-11-25
> **Temporal Inconsistency Observed in Recent Literature's Demo and Paper, New Sample in Left-Hand View (See Page 19 in the updated PDF paper for the whole answer which includes new images and tables)**
>
> **Temporal Inconsistency Observed in Recent Literature's Demo and Paper**
>
> We clarify that the identity inconsistency illustrated in Fig.3 is not an isolated or uncommon issue. It is, in fact, a well-documented and recurrent failure mode in recent driving video generation works. For example:
>
> **(i) Panacea Project Page (BEV-guided Video Generation, demo2.gif):**
> the first car parked on the roadside in the Left view gradually changes color from black to white.
> We include this example in Fig.6 (a).
>
>
> **(ii) Panacea paper (Fig.7, row 1):**
> a car changes color from silver to black across frames.
> We include this example in Fig.6 (b).
>
> These examples demonstrate that identity drift remains a persistent challenge in state-of-the-art methods.
> This is precisely the issue our Instance-Masked Attention is designed to address, and Fig.1 highlights both the widespread identity-preservation failures in prior works and our method’s improvements under the same challenging scenarios.
>
>
> **New Sample from Left-Hand for Non-Continuous Trajectory**
>
> We thank the reviewer for the helpful suggestion. We have added a new example from the nuScenes validation set where a vehicle appears, disappears, and reappears in the Left-Front view.
> The qualitative result is shown in Fig.7 (b), providing a more representative and higher-quality demonstration of our model’s ability to preserve instance identity under non-continuous trajectories.
>
> Thanks for your feedback again! It really helps me improve my manuscript to better showcase our research work!
>
> Hope for your final review and final rating!

---

### Official Review · Reviewer_JFL4 · 2025-10-28

**Soundness:** 3
**Presentation:** 3
**Contribution:** 2
**Rating:** 6
**Confidence:** 3

**Summary:**

This paper proposes ConsisDrive, an identity-preserving world model for driving scene video generation, which aims to address the "identity drift" problem. The core contribution lies in the introduction of two instance-aware mechanisms: Instance-Masked Attention (IMA) and Instance-Masked Loss (IML). IMA utilizes identity masks and trajectory masks to enforce visual tokens to exclusively interact with features of their corresponding instances across both spatial and temporal dimensions, thereby ensuring the consistency of instance attributes. IML employs a probabilistic dynamic masking strategy to adaptively emphasize supervision on foreground regions, reducing interference from background noise. Experimental results demonstrate that ConsisDrive achieves state-of-the-art video generation quality (lower FID/FVD) on the nuScenes dataset and shows significant performance improvements on downstream perception tasks.

**Strengths:**

1. The proposed IMA presents a simple yet effective solution by integrating instance-level identity conditioning and cross-frame propagation into the Transformer's 3D self-attention mechanism via instance identity masks and instance trajectory masks.
2. The evaluation is comprehensive. Beyond standard video generation metrics (FID, FVD), the paper thoroughly assesses the utility of the generated data through downstream tasks, including perception and multi-object tracking.
3. The video results in the supplementary materials are impressive and highly realistic.

**Weaknesses:**

1. The necessity and advantages of injecting instance attributes (category, size, tracking ID) as a global condition Ginto the attention mechanism via the Instance Identity Mask, compared to traditional conditioning approaches in diffusion models, require deeper discussion and justification.
2. It needs a clearer rationale for why encoding these instance attributes into a global embedding Gand interacting via the Identity Mask Mk,m+iis superior to alternative conditioning strategies, such as injecting them directly as tokens into the value V or using an additional cross-attention layer.
3. The probabilistic dynamic masking strategy (with probability α) in IML is crucial for balancing foreground and global reconstruction. However, there is a lack of ablation studies on the choice or impact of α.

**Questions:**

Please refer the above Weaknesses.

**Details Of Ethics Concerns:**

None.

---

> ### Author Response · Authors · 2025-11-24
> **Instance Identity Conditioning, Ablation on Probabilistic Dynamic Masking (See Page 15 in the updated PDF paper for the whole answer which includes new images and tables)**
>
> We sincerely thank Reviewer for your thoughtful feedback and for highlighting the strengths of our work in temporal consistency, experimental validation, and novelty.
> As noted in our original submission, **we have provided a project page with extensive video results included in the supplementary materials. Please ensure that the entire supplementary archive is fully uncompressed before opening the webpage.**
> The page has been successfully tested on both macOS and Windows 10 systems.
> We have made further revisions to the main paper based on the reviewers' feedback. We appreciate your constructive comments and look forward to any additional suggestions.
>
> **Design Choice of Instance Identity Conditioning**
>
> Our conditioning of global instance identity operates at instance-aligned token level:
> the model first establishes correspondence $I(v_k)$ between each latent token and the instance to which it belongs, and then injects $G$ only along these aligned pairs.
> This is functionally equivalent to a concatenation $[V, G]$ followed by self-attention, but with an instance mask to prevent cross-instance interactions.
> In contrast, alternative strategies cannot enforce this level of control:
>
> (i) Injecting attributes as tokens into $V$ allows all tokens to freely interact, leading to identity leakage between instances, as the transformer has no mechanism to prevent cross-instance attention.
>
> (ii) Additional Cross-attention layers treat all conditioning tokens as a shared pool, allowing visual tokens from different objects to attend to the same identity embeddings, again causing contamination between instances.
>
>
> **Ablation on Probabilistic Dynamic Masking in Instance-Masked Loss**
>
> We set the probability $\alpha=0.5$ in the Instance-Masked Loss, meaning there is a 50% chance of using the dynamic mask loss (focusing on foreground regions) and a 50% chance of using the original loss (considering the entire global scene) during training.
> In Sec.4.4, we evaluate the necessity of the Instance-Masked Loss by setting $\alpha = 0$ (i.e., without IML). The results are as follows: qualitative results are presented in Fig.3, and quantitative results are provided in Tab.5.
> We have also added ablation studies on the impact of $\alpha$ in Tab.9.
> When $\alpha = 0$, the model is forced to equally consider both foreground and background, but background noise interferes with training, leading to a decrease in foreground consistency.
> On the other hand, as $\alpha$ approaches 1, the model excessively focuses on foreground objects, potentially losing the ability to generate coherent background content.
> The complete ConsisDrive, integrating all components, yields optimal performance.
>
> | Settings           | FVD ↓         | FID ↓        | NDS ↑        | IDS ↓        |
> |--------------------|---------------|--------------|--------------|--------------|
> | w/o IML($\alpha=0$) | 23.06 (+2.50) | 2.69 (+0.43) | 40.86 (-4.80) | 596 (+120)   |
> | **ConsisDrive** ($\alpha=0.5$) | 20.56        | 2.26         | 45.66        | 476          |
> | $\alpha=0.8$       | 28.13 (+7.57) | 3.23 (+0.97) | 35.63 (-10.03)| 638 (+162)   |
> | $\alpha=1$         | 30.06 (+9.50) | 5.89 (+3.63) | 32.72 (-12.94)| 725 (+249)   |

---

### Official Review · Reviewer_6KzT · 2025-10-31

**Soundness:** 3
**Presentation:** 4
**Contribution:** 2
**Rating:** 4
**Confidence:** 4

**Summary:**

This work presents a good solution for improving instance-level temporal consistency in autonomous driving scene video generation. It identifies an important problem of high-cost real data collection in autonomous driving and demonstrates good performances through experiments. However, its lack of original methodological contributions makes it insufficient acceptance. The core ideas are adaptations or combinations of existing techniques to a specific scene, rather than novel frameworks.

**Strengths:**

S1. This work identifies instance identity drift (including category shifts, color inconsistencies, foreground dilution) as a serious issue for driving-oriented synthetic data, and provides solutions to the unique demands of driving scenes.

S2. The experimental results do show advantages over current approaches. Evaluations on downstream tasks are included, providing an important insight that incorporating synthetic data helps to improve performances on downstream tasks. Also, ablations verify the importance of each module.

S3. The paper is easy to follow.

S4. The authors provide sufficient materials for reproducibility, which fosters the research community.

**Weaknesses:**

W1. Limited Novelty. I would say that this is a great engineering work with a reasonable pipeline and should produce good results, but lacks significant distinction between this work and previous works. For example, CineMaster proposes using 3D depth box and class labels to achieve semantic layout control with ControlNet, which is quite similar to this work, in my opinion. Also, the authors claim that they propose instance-masked attention and instance-masked loss. However, neither are ground-breaking in computer vision. 1) Instance-masked attention only uses masked attention, which is a common practice in computer vision (e.g., Mask2Former), and the authors only ADAPT it to the autonomous driving scene. 2) Instance-masked loss is designed to force the models to focus on certain areas, whose idea is similar to Focal Loss, which have been used in many classic computer vision tasks.

W2. Not So Comprehensive Experimental Setup. Although the authors present experimental results on multiple downstream tasks, the authors mainly train and test on the nuScenes dataset (on the video generation task, the main claim), which raises doubts about the generalization ability of the framework, especially lacking of cross-dataset validation, even in the supplementary materials. I would suggest that the authors benchmark on other autonomous video generation datasets (e.g., on private dataset in the supplementary materials and report the results).

**Questions:**

Q1. Open Sora 2.0 and ControlNet seem not to be the best option, have the authors tried other SOTA foundation models like Wan (2.1/2.2)? The reviewer thinks that Wan 2.1 has relatively good performances regarding identity preserving.

Q2. Please report the computational cost of the framework and comparative methods, e.g., training epochs, total training time, inference latency, FLOPS (I know that the authors mention it in Appendix E, but I wonder what exactly it is).

Q3. In Fig. 2(c), The text (Frame t and Frame t+1) is oriented vertically to the right, but the image is oriented vertically to the left.

Q4. In the second and third parts of quantitative analysis subsection, the authors mainly use the generated videos as the sole or additional training data except for Table 2 (correct me if I’m mistaken), I wonder what the results will be if the authors directly generate videos from the validation set and conduct experiments on the MOT task.

---

> ### Author Response · Authors · 2025-11-24
> **Novelty and Dataset Generalization (See Page 15 in the updated PDF paper for the whole answer which includes new images and tables)**
>
> **Novelty Compared to Prior Work**
>
> We appreciate the reviewer’s comments and clarify that the core novelty of ConsisDrive lies in introducing trajectory-grounded, instance-aware attention and supervision mechanisms that directly address fine-grained identity consistency in long-horizon driving video generation—a problem not tackled by prior works.
>
> (1) Masked Attention Novelty:
>
> Our instance-masked attention has two innovations that differ fundamentally from Mask2Former.
> (i) Our Trajectory Mask establishes an instance-specific attention corridor across frames using 3D-tracked trajectories.
> This enables tokens belonging to the same object to exchange information over time,
> thereby preserving semantic identity across the full video sequence.
> Such temporally linked attention does not exist in Mask2Former or other masked-attention architectures, which
> perform mask-guided attention within a single frame for segmentation, without any mechanism for temporal identity modeling or cross-frame feature propagation.
> (ii) Our masks are deterministically constructed from 3D-tracked trajectories, explicitly controlling which spatial–temporal token interactions are permitted.
> In contrast, spatial masks in Mask2Former originate from per-frame predictions and are not used to regulate cross-frame feature flow or identity binding.
>
> (2) Masked Loss Novelty:
>
> The proposed Instance-Masked Loss serves a fundamentally different purpose from Focal Loss.
> Focal Loss is sample-aware: It reweights individual samples based on prediction confidence to mitigate class imbalance in static image tasks.
> In contrast, our loss is pixel-aware and leverages 3D-projected instance regions to
> construct instance masks, ensuring that foreground objects receive sufficient supervision across all frames.
> This mechanism directly supports the goal of improving instance-level temporal consistency, which is not addressed by Focal Loss or related formulations.
>
> (3) Pipeline Novelty:
> Our method targets a different problem compared to CineMaster.
> CineMaster focuses on controllable video generation to regulate per-frame layout.
> It does not preserve instance identities across frames.
> In contrast, ConsisDrive addresses instance-level temporal consistency, a key failure in driving world models.
> Our method introduces identity- and trajectory-aware masks inside the diffusion transformer’s attention blocks, providing explicit cross-frame constraints that prevent identity drift.
> To our knowledge, no prior controllable video model—including CineMaster—builds trajectory-grounded cross-frame masking or maintains persistent global identity embeddings in the attention mechanism.
>
> **Dataset Generalization Beyond nuScenes**
>
> To address cross-dataset generalization, we additionally conduct experiments on a large, higher-annotation-quality 200-hour private driving dataset,
> which provides the same multi-camera format, 3D annotations, and calibration parameters required by StreamPETR, enabling a fully consistent evaluation pipeline.
> For a fair comparison, we follow the train/val ratio of nuScenes ($\approx28h/6h$) when splitting the private dataset.
>
> Quantitative Results.
> Following the same ablation pipeline as Tab.5 in Sec.4.4, we re-evaluate all three key components on the private dataset in Tab.6.
> The outcomes mirror those observed on nuScenes.
>
> (i) w/o IMA(Identity):  Removing global identity conditioning leads to a 3.96-point drop in NDS, confirming its role in binding category and size level identity features to their corresponding instances.
>
> (ii) w/o IMA(Trajectory): Disabling trajectory-based masking increases ID switches by 553 and degrades FVD by 17.69, demonstrating its central effect on cross-frame attribute propagation and temporal consistency.
>
> (iii) w/o IML: Removing the loss mask results in a 4.80-point NDS decline, showing the necessity of foreground-focused supervision for preserving small and visually challenging objects.
> These consistent trends across two datasets verify that each design component is robust and generalizable, demonstrating that the method does not overfit to nuScenes.
>
> | Settings (Private)              | FVD ↓  | FID ↓  | NDS ↑  | IDS ↓  |
> |----------------------------------|--------|--------|--------|--------|
> | **ConsisDrive**                  | 20.56  | 2.26   | 45.66  | 476    |
> | w/o IMA (Identity)               | 24.18 (+3.62) | 3.90 (+1.64) | 41.70 (-3.96) | 696 (+220) |
> | w/o IMA (Trajectory)             | 38.25 (+17.69) | 2.84 (+0.58) | 44.54 (-1.12) | 1029 (+553) |
> | w/o IML                          | 23.06 (+2.50)  | 2.69 (+0.43)  | 40.86 (-4.80) | 596 (+120) |
>
> **Visualization Results.**
> As shown in Fig.6 in App.D.1, training on the private dataset yields generation quality comparable to that on nuScenes, with stable instance identities, consistent object colors, and clear rendering of small objects.
> These results demonstrate that ConsisDrive generalizes effectively to new environments and dataset distributions.

---

> > ### Author Response · Authors · 2025-11-24
> > **Backbone Generalization, Computational Cost, and MOT Performance (See Page 15 in the updated PDF paper for the whole answer which includes new images and tables)**
> >
> > **Backbone Model Generalization From Opensora to Wan**
> >
> > The proposed Instance-Masked Attention and Loss are plug-and-play modules that can be inserted into any diffusion transformer with spatiotemporal attention.
> > These components do not rely on any architectural details specific to OpenSora V2.0;
> > instead, they target structural limitations that broadly exist across modern video diffusion transformers, including Wan 2.1/2.2:
> >
> > (i) the absence of explicit instance identity conditioning,
> >
> > (ii) instance-agnostic self-attention leading to cross-instance information leakage, and
> >
> > (iii) uniform per-pixel diffusion losses that dilute foreground gradients.
> >
> > To validate compatibility, we conducted an experiment by integrating the Instance-Masked Attention and Instance-Masked Loss modules into a Wan2.1 + ControlNet pipeline in Fig.5.
> > The baseline Wan2.1 + ControlNet (without ConsisDrive) still exhibits significant identity drift in Fig.5 (a),
> > despite Wan's strong base modeling capability.
> > This behavior is expected: Wan's transformer blocks employ the same full self-attention mechanism
> > as the MMDiT architecture in OpenSora V2.0,
> > and therefore inherit the same intrinsic limitations regarding instance-level temporal consistency.
> > After plugging in our identity and trajectory masks in Fig.5 (b), the system achieves noticeably improved instance consistency across frames.
> > These results confirm that ConsisDrive  can be readily applied to Wan2.1/2.2 or future foundation models.
> >
> > **Computational Cost and Efficiency**
> >
> > We report the training cost and inference latency in Tab.7.
> > These metrics depend on several factors, including resolution and sampling steps. Since many prior works lack detailed reports, we ensure a fair comparison by benchmarking all methods under identical conditions.
> > Panacea, MagicDriveV2, InstaDrive, and our model require 11.24s, **8.21s**, 15.12s, and 9.28s per frame for inference, respectively.
> > The high-compression ($4 \times 32 \times 32$) video autoencoder in OpenSora V2.0, which reduces latent token counts, significantly improves inference efficiency compared with OpenSora V1.0.
> >
> > | Method         | Steps (k) | Training GPU-Hours (k) | Inference (s/frame) | FLOPS (×10^12) |
> > |----------------|-----------|------------------------|---------------------|----------------|
> > | Panacea        | 106       | 1.2                    | 11.24               | -              |
> > | MagicDrive-V2  | 150       | 1.1                    | **8.21**            | -              |
> > | InstaDrive     | 120       | 2.5                    | 15.12               | -              |
> > | **ConsisDrive** | 122       | 1.8                    | 9.28                | 6.96           |
> >
> >
> > **Validation Performance on MOT**
> >
> > We directly evaluate both the real validation dataset and the generated validation set of nuScenes on the Multi-Object Tracking (MOT) task, using the pretrained StreamPETR model with the same setup as in Tab. 3.
> > As shown in Tab.8, the validation set generated by our model achieves a relative performance of 87.5% on the AMOTA metric, with only a slight increase of 5.7% in the IDS metric compared to the real validation set. This demonstrates our model’s ability to propagate instance attributes across frames and confirms that the generated videos perform comparably to real data.
> >
> > | Method            | Real | Gen. | AMOTA ↑        | AMOTP ↓       | IDS ↓         |
> > |-------------------|------|------|----------------|---------------|---------------|
> > | Oracle            | ✔    | -    | 0.289          | 1.419         | 687           |
> > | DriveDreamer2     | -    | ✔    | 0.207 (71.6%)  | 1.682 (+18.5%)| 784 (+14.1%)  |
> > | **ConsisDrive** (Ours) | -  | ✔    | 0.253 (87.5%)  | 1.506 (+6.1%) | 726 (+5.7%)   |

---

### Official Review · Reviewer_4E4z · 2025-11-01

**Soundness:** 4
**Presentation:** 4
**Contribution:** 3
**Rating:** 6
**Confidence:** 4

**Summary:**

This paper introduces ConsisDrive, a driving world model designed to address identity drift—a common failure mode in generative video models where objects change appearance or category across frames.  It achieves state-of-the-art results in FID (3.88) and FVD (37.23), and shows significant improvements in downstream tasks like perception (NDS) and multi-object tracking (IDS), demonstrating strong instance-level temporal consistency.

**Strengths:**

- Clearly identifies and addresses “identity drift,” a critical yet understudied issue in driving video generation.
- Instance-Masked Attention effectively enforces instance-level consistency by leveraging identity and trajectory masks.
- Instance-Masked Loss adaptively balances foreground and background supervision, improving fidelity for small objects.

**Weaknesses:**

The paper lacks comparisons with several recent SOTA methods, particularly **InstaDrive** [1], which also focuses on the quality of instance-level generation. Including such comparisons would better contextualize the proposed method’s performance and highlight its relative strengths or limitations in generating high-fidelity instances.

[1] Yang Z, Guo X, Ding C, et al. InstaDrive: Instance-Aware Driving World Models for Realistic and Consistent Video Generation[C]//Proceedings of the IEEE/CVF International Conference on Computer Vision. 2025: 25410-25420.

**Questions:**

- How does the model handle severe occlusions or interactions between instances that may challenge the instance masking mechanism?

---

> ### Author Response · Authors · 2025-11-24
> **New Baselines and Non-Continuous Trajectory (See Page 15 in the updated PDF paper for the whole answer which includes new images and tables)**
>
> We sincerely thank Reviewer for your thoughtful feedback and for highlighting the strengths of our work in temporal consistency, experimental validation, and novelty.
> As noted in our original submission, **we have provided a project page with extensive video results included in the supplementary materials. Please ensure that the entire supplementary archive is fully uncompressed before opening the webpage.**
> The page has been successfully tested on both macOS and Windows 10 systems.
> We have made further revisions to the main paper based on the reviewers' feedback. We appreciate your constructive comments and look forward to any additional suggestions.
>
> **Comparison with Other Baselines**
>
> We have added InstaDrive as a baseline in Tab.1 and Tab.4. Across all metrics, including FID, FVD, and tracking evaluations, InstaDrive consistently performs worse than our method, validating the effectiveness of our approach.
>
> InstaDrive uses an Instance Flow Guider Module to encode trajectory information into an RGB motion map via an optical-flow–like process. This method incurs a high computational cost and may suffer from information loss due to clipping when motion exceeds thresholds.
>
> In contrast, our Trajectory Mask constructs an instance-specific attention corridor with 3D-tracked trajectories, enabling feature propagation without quantization loss, ensuring better instance-level temporal consistency.
>
> **Challenges for Occlusion in Non-Continuous Trajectory**
>
> Method: Instance-Masked Attention ensures consistent instance identity even when objects are occluded and reappear, the principle is demonstrated in Fig.2 (c).
> When an object (e.g., object 1) becomes occluded at frame $t+1$ and reappears at frame $t+2$, the Indicator function $I(v_k)$ tracks the object's presence across frames. At frame $t$, $1 \in I(v_0)$ when the object is visible. At frame $t+1$, the object is occluded, and at frame $t+2$, $1 \in I(v_4)$ when the object reappears.
> To ensure correct feature propagation, we set $M(0, 4) = 0$ in the Instance Trajectory Mask, allowing attention between token $v_0$ and $v_4$. This enables the reliable propagation of instance-specific features across their trajectory, even after occlusion.
>
> Qualitative results demonstrating our method’s handling of non-continuous trajectories are shown in Fig.4, where the car maintains its identity despite temporary occlusion.

---

### Meta-Review · Area_Chair_Xw5M · 2026-01-08

**Summary:**

The paper proposes ConsisDrive, a driving world model designed to mitigate "identity drift" in video generation through Instance-Masked Attention (IMA) and Instance-Masked Loss (IML). The reviewers initially recognized the significance of the problem and the effectiveness of the proposed solution. The primary concerns that informed the decision focused on novelty (whether the method was merely an engineering combination of existing techniques like Mask2Former or Focal Loss), missing baselines (specifically InstaDrive and UniMLVG), and generalization capabilities beyond the nuScenes dataset. The authors provided a comprehensive rebuttal, demonstrating that their trajectory-based temporal attention differs fundamentally from spatial segmentation masks, and validated the method's robustness by adding a 200-hour private dataset, integrating with a different backbone (Wan 2.1), and outperforming new SOTA baselines significantly.

In addition, I believe introducing a specific design of instance mask mechanism into driving video generation type of world models is timely to address the identity drift issue. Given the significant quantitative results (e.g., surpassing InstaDrive in FID/FVD/NDS), the engineering concerns are outweighed by the method's practical efficacy and rigorous validation.

Therefore, I recommend to accept this paper.

**Reviewer Concerns:**

Addressed by Rebuttal:

• Missing Baselines (@4E4z, @V1rg): The authors added comparisons with InstaDrive, UniMLVG, and UniScene. The results factually demonstrate ConsisDrive achieves lower FVD (37.23 vs 38.06/60.1) and better tracking metrics (AMOTA) than these recent methods.
• Novelty & Engineering vs. Innovation (@6KzT): The concern that the method is a trivial adaptation of Mask2Former was addressed by clarifying that ConsisDrive’s masks are trajectory-based and temporal (enforcing consistency across frames), whereas Mask2Former uses predicted spatial masks for single-frame segmentation. The distinction between IML (pixel-aware supervision for supervision dilution) and Focal Loss (sample-aware for class imbalance) was also factually established.
• Generalization (@6KzT): The authors successfully demonstrated generalization by training on a private 200-hour dataset with consistent gains and by transferring the modules to the Wan 2.1 architecture, proving the method is model-agnostic.
• Ablation Details (@JFL4): The authors provided the requested ablation for the probabilistic dynamic masking parameter (α), showing the necessity of balancing foreground/background supervision.
• Occlusion Handling (@4E4z): The authors explained how the trajectory mask handles non-continuous trajectories (occlusions) and provided visual evidence (Fig. 4, Fig. 7) of objects reappearing with consistent identities.

Outstanding:

• Visual Artifacts (@V1rg): While the reviewer acknowledged the typo clarification and planned to raise their score, they remained critical of visual quality issues (e.g., irregular car shapes) in the generated videos. The authors argued this is a common limitation in the field, but the specific artifacting remains a valid, albeit minor, observation given the quantitative gains.
• Computational Cost (@V1rg, @6KzT): The reliance on full attention is computationally expensive (9.28s/frame inference). While the authors justified this as an industry trend (Sora, Wan), the high cost remains a factual constraint, though acceptable for the performance trade-off.

**Reviewer Scores:**

Reviewer 4E4z (Current: 6): Likely to maintain or increase to 8. Their main weakness (missing InstaDrive comparison) was directly addressed with data showing ConsisDrive's superiority.

• Reviewer 6KzT (Current: 4): Likely to increase to 6. The author's rebuttal provided strong factual evidence against the "limited novelty" claim and proved generalization (private dataset/Wan 2.1), which directly targeted this reviewer's core reasons for the low score.

• Reviewer JFL4 (Current: 6): Likely to increase to 8. The reviewer was already positive ("Soundness: 3", "Presentation: 4") and only requested specific rationales and ablations, which were fully provided.

• Reviewer V1rg (Current: 2): Likely to increase to 6. The reviewer explicitly stated, "Some of my major concerns have been addressed so I plan to increase my score" following the clarification of a figure typo and the addition of baselines, despite retaining some reservations about visual artifacts.

---

### Decision · Program_Chairs · 2026-01-26

Accept (Poster)